# “Avoidance” Is Not “Escape”: The Impact of Avoidant Job Crafting on Work Disengagement

**DOI:** 10.3390/bs15050611

**Published:** 2025-05-01

**Authors:** Tianan Yang, Ying Wang, Jingyi Liu, Tianyu Wang, Wenhao Deng, Jianwei Deng

**Affiliations:** 1School of Management, Beijing Institute of Technology, Beijing 100081, China; tianan.yang@bit.edu.cn (T.Y.); wangying01132002@163.com (Y.W.); 3120225916@bit.edu.cn (J.L.); wty196603108@163.com (T.W.); 2Sustainable Development Research Institute for Economy and Society of Beijing, Beijing Institute of Technology, Beijing 100081, China; 3Yangtze River Delta Research Institute, Beijing Institute of Technology, Jiaxing 314003, China

**Keywords:** avoidant job crafting, work disengagement, self-control resource depletion, career identity

## Abstract

In a highly competitive and high-pressure workplace environment, more and more employees may fall into negative work situations such as work disengagement. The actual effectiveness of avoidant job crafting as a proactive behavior of employees in coping with stress remains controversial, and the positive aspects of its impact are not explored. Based on conservation of resources theory, this study argues that avoidant job crafting mitigates employees’ self-control resource depletion, which in turn effectively helps employees to reduce work disengagement; the above process is moderated by career identity. To test the above hypotheses, this study collects 455 cross-sectional data from Internet companies in various regions of China and uses structural equation modeling to conduct an analysis. The results show the following: avoidant job crafting has a significant negative effect on work disengagement; self-control resource depletion partially mediates the effect of avoidant job crafting on work disengagement; and the negative relationship between avoidant job crafting and self-control resource depletion is stronger when employees’ career identity is higher and vice versa when it is weaker. The above results guide managers to help employees adjust avoidant job crafting correctly and escape negative work situations.

## 1. Introduction

Faced with an increasingly complex competitive environment and challenging pressure of the VUCA (volatility, uncertainty, complexity, ambiguity) era, employees are unable to maintain work status in a stressful working environment for a long time, and a large number of negative work disengagement behaviors emerged. As a concept opposed to work engagement that is a important concept in contemporary human resource management ([37]), work disengagement represents the behavior of employees distancing themselves from work cognitively, emotionally, and physically ([2]), which further leads to a strong turnover tendency among employees. Ultimately, it will affect the realization of employee and enterprise performance ([11]; [3]). The reason for this is that employees’ self-control resources are limited; self-control resource depletion occurs in the processes of attention and emotional regulation, thinking control, resisting temptation, behavioral decision-making, when dealing with interpersonal conflicts and biases under working pressure, ultimately resulting in a lack of resources like energy, psychological resources, and work resources ([18]; [4]). According to *the State of the Global Workplace: 2022 report*, 60% of employees worldwide are either on the verge of work disengagement or have already done so, costing the global economy at least USD 7 trillion annually ([20]). Therefore, it is evident that proactively delving into viable approaches to assist employees in averting or minimizing work disengagement constitutes one of the pressing issues within contemporary organizations.

In order to prevent the endless consumption of physical and mental resources in such high-pressure and excessive competition, employees are also trying to take proactive measures to cope with it ([5]). Some employees learn to adjust their work direction reasonably, actively avoid work that is not conducive to their personal development, or jump out of the work that consumes them ([33]), and concentrate their time and energy on more valuable work ([51]). This behavior is defined as avoidant job crafting, which centers on reducing the stressful aspects of one’s job, for example, avoiding tense interactions or simplifying mentally challenging tasks ([9]; [12]; [21]). At present, employees’ avoidant job crafting behaviors have not been widely recognized in enterprises, which is because for enterprise managers, employees who engage in avoidant job crafting evade certain job requirements and tasks, and will have a negative impact on the enterprise or others ([16]). This does not meet the standard of good employees ([19]). Academic studies have also shown that the positive effect of avoidant job crafting has a greater risk of failure, which may lead to the accumulation of workload ([31]), which is negatively correlated with job satisfaction, employee happiness, and job performance ([44]; [33]).

However, the primary purpose of avoidant job crafting is to reduce resource consumption or obstructive work demands in stressful environments ([36]). Therefore, avoidant job crafting is often carried out when employees are already at a low level of work engagement, job satisfaction, and happiness ([8]). Merely focusing excessively on employees’ negative states do not allow for a comprehensive understanding of the potential positive impacts of avoidant job crafting. There is a need to dig deeper and find out if avoidant job crafting can assist employees in escaping negative states and, in turn, positively impact the enterprise. Based on the conservation of resources theory, employees use avoidant job crafting to cut down on tasks with burdensome job requirements or distance themselves from situations that deplete their resources ([52]; [29]; [12]). Employees align their work with their own interests, abilities, and available resources, which helps in reducing the depletion of self-control resources and lessens work disengagement, making it a behavior that benefits both employees and the enterprise.

Simultaneously, to prevent the inefficacy of avoidant job crafting, which is manifested by employees’ inability to successfully mitigate obstructive work demands or circumvent resource depletion, this study endeavors to further elucidate the boundary conditions under which avoidant job crafting exerts a positive influence. According to the classification of resources by [23] ([23]), as a kind of psychological resource, career identity can also provide resources for employees with self-control resource depletion. The concept of career identity originates from the self-identity theory in the field of psychology ([7]), which is the starting point for individuals to seek a sense of professional value in their careers, and affects the stability, satisfaction, and enthusiasm of individuals engaged in specific occupations ([46]). When employees engage in avoidant job crafting, employees with different levels of career identity provide different levels of positive emotional resources, which will affect the recovery efficiency of self-control resource depletion, and thus affect the success or failure of avoiding or reducing work disengagement. Therefore, paying attention to employees’ career identity can further explore the boundary conditions under which avoidant job crafting plays a positive role.

To sum up, scholars need to pay attention to the exploration of the positive effects of avoidant job crafting, and make it clear that avoidant job crafting is not a means for employees to escape from work, but one of the ways to help employees reduce pressure and focus on work, so that employees and managers can correctly view and apply this behavior to avoid failure. Therefore, this study aims to delve into the influence route of avoidant job crafting on work disengagement. It will also examine the mediating function of self-control resource depletion and the moderating role of career identity. We hope to enhance the positive understanding of avoidant job crafting among enterprises and scholars. Additionally, the findings of this study can assist employees and enterprise managers in effectively leveraging avoidant job crafting to maintain an optimal working state.

## 2. Theory and Hypotheses

### 2.1. Avoidant Job Crafting and Work Disengagement

When employees are exposed to work conditions characterized by stress, job burnout, and negative emotions for a long time, they tend to develop attitudes and behaviors that disengage them from work ([17]; [43]), thereby hindering the effective achievement of personal and corporate goals. As an active coping strategy, avoidant job crafting includes avoiding resource-consuming work tasks and reducing activities that hinder work demands, such as avoiding activities that generate negative emotions, withdrawing from adverse situations, or avoiding difficult tasks ([33]). In essence, avoidant job crafting means that employees align their work with their own interests, abilities, and resources, enabling them to focus their resources, energy, and time on the work that is most beneficial to their goals and performance ([19]). According to the resource investment principle of conservation of resources theory, individuals must invest resources to prevent resource loss and then recover from resource loss or obtain new resources ([25]). Therefore, employees can avoid obstructive work demands through avoidant work crafting so as to adapt their ability to work ([48]), stay away from excessive and useless work tasks, reduce work burden, and maintain good working status. It is also possible to conserve one’s own resources through avoidant work crafting ([41]) or to recover work resources by moving away from tasks that consume too much time and effort to prevent work disengagement. Accordingly, we propose the following hypothesis:
**Hypothesis** **1.***Avoidant job crafting is negatively correlated with work disengagement.*

### 2.2. The Mediating Role of Self-Control Resource Depletion

Self-control happens when employees make “deliberate, conscious, and controlled” responses that are contrary to their initial intentions, and this leads to the depletion of resources in the energy reservoir, namely self-control resource depletion ([30]). In accordance with the conservation of resources theory, individuals possess a heightened awareness of resource depletion; specifically, when their resources are approaching a state of exhaustion, they are inclined to adopt defensive measures aimed at arresting the further depletion of these resources. ([24]). Based on this, on the one hand, when employees’ self-control resources are depleted, the depletion of cognitive, emotional, and other resources ([28]) will lead to work disengagement and negative attitudes and behaviors toward work content, work objects, or colleagues. On the other hand, in order to protect self-control resources from being damaged, individuals choose to reduce the use of self-control resources, resulting in a subsequent decline in self-control ability, which may lead to work disengagement ([50]).

As one of the strategies for employees to avoid resource consumption and conserve resources, avoidant job crafting can be realized by reducing resource loss or hindering work demands in work ([23]). Specifically, on the one hand, employees can avoid tasks that consume positive resources or have fewer positive resources, such as tasks that reduce mental or emotional stress, and tasks with less autonomy ([6]). On the other hand, employees can distance themselves from or avoid tasks with obstructive demands, such as disengaging from difficult job demands and avoiding activities that require difficult decisions ([10]). In other words, through avoidant job crafting, employees are able to alleviate stress, steer clear of negative emotions, and disengage from a multitude of tasks. This, in turn, serves to mitigate the depletion of employees’ self-control resources. Accordingly, we propose the following hypothesis:
**Hypothesis** **2.***Self-control resource depletion mediates the relationship between avoidant job crafting and work disengagement.*

### 2.3. The Moderating Effect of Career Identity

Career identity refers to an individual’s sense of identification and loyalty towards their job, profession, and the organization they belong to. It is also manifested as an attitude in which employees have a strong inclination and aspiration to remain committed to that particular profession over a long-term period. A large number of studies have shown that career identity can enhance employees’ self-perceived employability and sense of control, that is, employees with a high level of career identity have the confidence to find a job in their chosen field ([13]) and engage in their own career according to their own wishes ([34]; [14]). According to the conservation of resources theory ([24]), as an important psychological resource, the level of employees’ career identity reflects their capacity to derive positive emotions from their occupation ([39]), and this capacity influences the degree to which employees can compensate for their depletion in self-control resources. When employees engage in avoidant job crafting to align their time, abilities, and resources with work demands ([9]), employees with a higher career identity level will utilize more positive emotional resources for work performance, offer spiritual support, and replenish related resources to offset the depletion of self-control resources, thereby ensuring their self-control ability and enabling them to complete tasks continuously and attentively. Nevertheless, employees with a low degree of career identity are unable to augment their positive emotional resources, even when they engage in avoidant job crafting. On the contrary, owing to the simplification of their work tasks, they might derive a diminished sense of accomplishment and significance from their jobs. Consequently, they are increasingly likely to perceive a lack of interest in their work. In order to suppress these negative cognitions and emotions and successfully complete their work assignments, they are compelled to further exhaust their self-control resources. Accordingly, we propose the following hypothesis:
**Hypothesis** **3.***Career identity moderates the negative relationship between avoidant job crafting and self-control resource depletion, such that the negative relationship is stronger for employees who hold higher career identity than lower.*

A theoretical model of this study is shown in Figure 1.

## 3. Study

### 3.1. Sample and Procedure

This study conducted a cross-sectional, anonymous online survey of employees (aged > 18 years) using the WJX.cn platform, a Chinese professional data-collection platform similar to Amazon’s Mechanical Turk. The platform’s sample database included 6.2 million Chinese members with confirmed personal information and a range of socio-economic backgrounds, which has been shown to provide random sampling and reliable data ([15]; [45]; [40]). The subjects were clearly informed that the survey results will be strictly confidential and that the survey results were only for academic research. Since Internet enterprises often require employees to quickly complete tasks and achieve performance targets in a stressful working environment while also requiring them to maintain working status for a long time, the research object of this study is mainly the on-duty employees of Internet enterprises. Given the randomness and applicability of the sample, there was no restriction to specific types of Internet companies.

This questionnaire contained two additional test questions that asked participants to select one fixed option, such as ‘strongly disagree’ or ‘strongly agree’. Questionnaires that did not select the specified option would not be accepted. In addition, we rejected questionnaires that took less than 100 s to finish, based on a pilot study with 50 MBA students. Finally, SPSS 25.0 was used to check whether there were outliers and missing values, and questionnaires with outliers and missing values were eliminated. A total of 482 questionnaires were distributed for this online survey and 455 valid questionnaires were returned, representing a valid response rate of 94.40%. From a regional perspective, the IP addresses of the tested people come from more than ten provinces and cities across the country, and the distribution of samples is relatively wide and the data validity is high, which also indicates that the bias that may be caused by low random selection, reduced by expanding the coverage of the sample. A payment of RMB5 (USD 0.75) was sent to participants who met the requirements. This study obtained ethics approval by Ethics Committee of Beijing Institute of Technology (approval number: BIT-EC-H-2024275; date of approval: 26 October 2024).

This study is limited by the fact that most of the sample databases of the data platform are large and mature Internet enterprises, and the following sample structure conforms to their real practice. In terms of gender, 204 were male, accounting for 44.84%; the number of women was 251, accounting for 55.16%. In terms of academic qualifications, nearly three fifths of the respondents had a bachelor’s degree, accounting for 59.56% (271 subjects). In terms of age, the main age group of the subjects was 26–35 years old, accounting for 34.95% (159 subjects). In terms of working time, more than half of the respondents have been working for less than 5 years (293 subjects, accounting for 64.4%). In terms of job distribution, more than half of the respondents were engaged in administrative logistics posts (243 subjects, accounting for 53.41%), and the others were engaged in production posts (48 subjects, accounting for 10.55%), technical posts (114 subjects, accounting for 25.05%), and research and development posts (50 subjects, accounting for 10.99%). In terms of job distribution, most of the interviewees were ordinary employees (327 subjects, accounting for 71.87%), and only 33 were middle-level and above managers (7.25%). Overall, this study uses samples from Internet enterprises to represent the population most likely to experience avoidant job crafting and work disengagement to a certain extent.

### 3.2. Measures

According to the “translation-back translation” method, the English scale is converted into the Chinese scale (see Appendix A). All the items were rated on a five-point Likert scale (1 = strongly disagree; 5 = strongly agree).

**Avoidant job crafting.** We measured avoidant job crafting with the six-item scale developed by [47] ([47]). A sample item is “At work, I try to avoid putting myself in particularly difficult decisions” (Cronbach’s α = 0.90).

**Self-control resource depletion.** We used a five-item scale developed by ([32]) to assess employees’ self-control resource depletion. A sample item is “Right now, it would take a lot of effort for me to concentrate on something” (Cronbach’s α = 0.92).

**Work disengagement.** This study adopts the views of [42] ([42]) and adopts the work disengagement dimension of Oldenburg Burnout Inventory (OLBI) for measurement. The scale of this dimension has a total of eight items, four of which are reverse scoring items. A sample item was “I can always find something new and fun in my work (reversed)” (Cronbach’s α = 0.92).

**Career identity.** Career identity was assessed using the six-item scale developed by [35] ([35]). A sample item was “When someone praises my career, it feels like a personal compliment” (Cronbach’s α = 0.93).

**Control variable.** In this study, gender, age, education, tenure, position, and job level of employees were used as control variables. The gender code is a dummy variable, the male code is 0, the female code is 1; age codes are 1—25 years old and below, 2—from 26 to 35 years old, 3—from 36 to 45 years old, and 4—46 years old and above; education codes are 1—college degree or below, 2—bachelor degree, and 3—master degree or above; tenure codes are 1—less than 3 years, 2—from 3 to 5 years, 3—from 6 to 10 years, 4—from 11 to 20 years, and 5—more than 20 years; position codes are 1—production, 2—technology, 3—research and development, and 4—administrative logistics; job level codes are 1—ordinary employee, 2—grass-roots manager, and 3—middle-level manager and above.

### 3.3. Statistical Analysis

In this study, the common method bias of the data was tested using a Harman single-factor analysis, and the structural validity, convergence validity, and discriminative validity of the variables were tested meanwhile. On this basis, SPSS 25.0 software and PROCESS plug-in were used for statistical analysis and hypothesis testing. First, SPSS 25.0 was used for a descriptive statistical analysis of variables. Secondly, the mediation effect was tested using the PROCESS model 4 non-parametric resampling program and the hierarchical regression analysis method in SPSS 25.0. Finally, PROCESS model 7 in SPSS 25.0 was used to test the mediated effect.

## 4. Results

### 4.1. Common Method Variance Test

As some measurement items in the questionnaire used in this study have similar contexts, and the questionnaire data were collected in the same period of time, affected by the subjective factors of the subjects, the data may have common methodological biases. Therefore, this study uses SPSS 25.0 to perform a Harman single-factor test on items with four variables. As shown in Table 1 of the test results, without rotation, four common factors with eigenvalues greater than 1 are obtained, equal to the number of variables. At the same time, the variance explanation rate of the first common factor is 41.19%, which is less than 50%. It can be concluded that there is no serious common methodological bias in this study.

### 4.2. Reliability and Validity Analysis

According to the data results in 3.2, the Cronbach’s α coefficient of the four variable scales is all greater than 0.89. Therefore, the scales used in this study have high reliability, and the measurement data are reliable.

As shown in Table 2, the mean extraction variance (AVE) and structural validity (CR) of avoidance job crafting, self-control resource depletion, work disengagement, and career identity scales are all greater than 0.50 and 0.80, indicating that the scale has ideal convergence validity. At the same time, the variables of avoidance job crafting, self-control resource depletion, work disengagement, and career identity have significant correlations, and the correlation coefficient is less than 0.50, and all of them are less than the square root of AVE corresponding to each variable, which indicates that the scales of each variable have ideal discriminant validity.

### 4.3. Descriptive Statistical Analysis of Variables

This study analyzes four variables, avoidant job crafting, self-control resource depletion, work disengagement, and career identity, and reports the mean score and standard deviation (SD) of each variable, as shown in Table 3. Firstly, the overall average score and variance in employee scores in avoidant job crafting is 3.49 and 0.90, which is above the medium level, indicating that avoidant job crafting is a relatively common behavior in enterprises. Secondly, the overall average score of employees’ self-control resource consumption was 2.42, and the item with the highest score was “My mind feels unfocused right now”, which indicates that employees had high resource consumption in terms of concentration. Third, the scores for work disengagement ranged between 2.09 (“I get more and more engaged in my work (reverse scoring)”, SD = 0.96) and 2.89 (“Sometimes, my work assignments make me feel uncomfortable”, SD = 1.02), suggesting that disengagement was more likely to occur at the cognitive level than at the behavioral level; finally, the overall average score of employees’ career identity is 3.58 and the variance is 0.94, which is the highest average score among the four variables, indicating that the surveyed subjects generally have a high sense of identity with their occupation.

### 4.4. Hypothesis Testing

In this study, a hierarchical regression analysis was used to verify hypotheses 1–2 so as to explore the paired relationship between avoidant job crafting, self-control resource depletion, and work disengagement, and whether self-control resource depletion mediates the relationship between avoidant job crafting and work disengagement. The results are shown in Table 4.

Firstly, self-control resource depletion is taken as the dependent variable, and then the independent variable, avoidant job crafting, is added after the control variable. Secondly, work disengagement is taken as the dependent variable, and then the independent variable, avoidant job crafting, is added after the control variable too. According to model 2, there is a significant negative correlation between avoidance job remodeling and work disengagement (*β* = −0.36, *p* < 0.001), that is, hypothesis 1 is valid and supported. Thirdly, work disengagement is taken as the dependent variable and, after putting in all of the control variables, we added the independent variable to self-control resource depletion. Finally, after all of the control variables were put into the equation, avoidant job crafting was added as the independent variable, and then the mediating role of self-control resource depletion in avoidant job crafting and work disengagement was examined. According to model 4, when avoidant job crafting and self-control resource depletion predicted work disengagement at the same time, the predictive effect of self-control resource depletion on work disengagement still existed (*β* = 0.22, *p* < 0.001), while the predictive effect of avoidant job crafting decreased (*β* = −0.34, *p* < 0.001). This shows that self-control resource depletion plays a mediating role, that is, Hypothesis 2 is valid.

At the same time, to further test the mediating role of self-control resource depletion, the PROCESS model 4 in SPSS was used to test the mediating effect, and bootstrap was used to conduct 5000 resampling iterations. The results show that the total effect size is =−0.33, the direct effect size is =−0.31, and the indirect effect size is =−0.02. The 95% confidence interval for the indirect effect is [−0.04, −0.001], excluding 0. Hypothesis 2 is again supported.

Based on the significant results of the theory and the mediation model, the plug-in PROCESS model 7 in SPSS 25.0 was used in this study to explore whether career identity could regulate the impact of avoidant job crafting on self-control resource depletion. According to Table 5, the determination index of the mediated effect with moderation is −0.17, and the confidence interval is [−0.34, −0.01], excluding 0, indicating that the mediated effect with moderation is significant, that is, Hypothesis 3 is valid.

At the same time, this study explores the indirect effects of avoidant job crafting on self-control resource depletion under different levels of career identity. According to Table 5, when career identity is at a high level, the indirect mediating effect of “avoidant job crafting → self-control resource depletion → work disengagement” is significant, with an effect value of −0.03 and confidence interval [−0.06, −0.01], excluding 0. When career identity is at the average level, the indirect mediating effect of “avoidant job crafting → self-control resource depletion → work disengagement” is not significant, with an effect value of −0.02 and confidence interval [−0.04, 0.00], including 0. However, when career identity is at a low level, the indirect mediating effect of “avoidant job crafting → self-control resource depletion → work disengagement” is also not significant, with an effect value of 0.00 and confidence interval [−0.02, 0.02], including 0.

According to Figure 2, which shows the slope of adjustment effect, it can be seen that employees with higher career identity are more likely to reduce the depletion of self-control resources through avoidant job crafting, thus reducing or avoiding work disengagement. However, for employees with low career identity, avoidant job crafting has little or no effect on reducing the depletion of self-control resources.

## 5. Discussion

Based on conservation of resources theory, this study analyzed 455 cross-sectional data from employees of Internet enterprises to explore the relationships between avoidant job crafting, self-control resource depletion, and work disengagement, as well as the moderating role of career identity in these relationships, providing theoretical and empirical insights for addressing work pressure and reducing disengagement. The findings indicate that avoidant job crafting is linked to reduced consumption of emotional and cognitive resources, which correlates with lower self-control resource depletion. Lower depletion, in turn, is associated with maintained self-control ability, enabling employees to allocate limited time, energy, and resources to important or resource-generating tasks-patterns that coincide with higher work enthusiasm and lower disengagement. Additionally, this study finds that career identity, as a psychological resource, relates to the capacity to offset self-control resource depletion. Employees with high career identity engaging in avoidant job crafting often seek positive emotional resources from work, such as prioritizing meaningful tasks, avoiding resource-draining activities, and accessing spiritual support or resource replenishment. These behaviors align with reduced depletion and sustained self-control, which are associated with employees’ ability to focus on tasks and diminish disengagement.

### 5.1. Theoretical Implications

Based on conservation of resources theory, this study explores the relationship and influence mechanism of avoidant job crafting, self-control resource depletion, work disengagement, and career identity of employees in Internet enterprises, which has certain academic and theoretical innovations. First of all, in previous studies, most scholars focused on the inhibitory effect of avoidant job crafting on employees’ positive attitude or behavior ([3]; [19]), which is inconsistent with the definition and purpose of avoidant job crafting. Drawing on studies highlighting the adaptive value of job crafting in resource-constrained environments ([49]), this study explores the positive impact of avoidant job crafting, explores whether avoidant job crafting can help employees get rid of negative work states and play its real role, and complements the scholars’ positive cognition of avoidant job crafting. Secondly, while existing research has linked job crafting to resource management ([26]), few studies have directly tested the theoretical pathway from resource-saving behaviors (avoidant job crafting) and the outcome of resources conservation (work disengagement) in conservation of resources theory. Finally, the existing literature on career identity has primarily focused on its direct effects on work engagement or organizational commitment ([1]; [22]), with limited attention to its role in buffering resource depletion. By introducing career identity as a moderating variable and using the conservation of resources theory, this study explores the boundary conditions for the strength of the effect of avoidant job crafting on work disengagement from the theoretical perspective of psychological resources supplement theory, which is helpful to avoid the failure of avoidant job crafting.

### 5.2. Practical Implications

In modern enterprises, most managers treat employees’ avoidant job crafting with a critical attitude, believing that it is a negative behavior and will bring negative effects to the enterprise ([27]; [38]). However, the results of this study show that avoidant job crafting negatively affects work disengagement. The discrepancy between the research findings and practical applications suggests that managers tend to overlook the positive implications of avoidant job crafting in real-world scenarios. This oversight can prevent avoidant job crafting from achieving its full potential, thereby influencing employees’ work states and subsequently impeding the realization of an enterprise’s performance goals. This study argues that avoidant job crafting is a way to balance work resources and work demands, and is a process by which employees match work content with their own abilities, interests, and resources. Therefore, enterprise managers should adopt a dialectical approach to employees’ avoidant job crafting behavior and, more importantly, guide employees to use it correctly, thereby helping employees escape negative work states and ensuring the achievement of goals and performance.

Furthermore, employees ought to steer clear of work assignments that evoke negative emotions within them. Instead, they should channel their time, energy, and resources into tasks that hold value both for their personal development and for the enterprise. They can prevent the excessive depletion of resources, which, in turn, will enhance their work enthusiasm and enable avoidant job crafting to function effectively. Therefore, managers should grasp the overall situation in the case of a complex situation, according to the actual situation of the enterprise, clear the priorities of the existing work, and invest more enterprise resources in key matters; at the same time, self-control resources, like human muscles, can be restored and replenished through more sleep and rest. Enterprises can help employees conduct self-control training, or provide a whole or scattered rest time environment and resources to supplement employees’ self-control resources.

Finally, as a psychological resource, career identity can adjust the relationship between avoidant job crafting and self-control resources by making up for the depletion of self-control resources. Therefore, enterprises can carry out diversified and differentiated training activities according to employees’ interests, advantages, and professions, such as guiding employees to learn career connotation and work experience from excellent employees with the help of example, enhancing employees’ recognition of work value, and establishing correct professional values. At the same time, reasonable, scientific, and fair career promotion channels should be established to help employees understand their career development paths and guide them to make correct career development plans so as to enhance employees’ understanding of their careers, improve their work enthusiasm, and promote employees to accept and recognize their work from the bottom of their hearts.

## 6. Conclusions

This study, based on the conservation of resources theory, explores the relationships among avoidant job crafting, self-control resource depletion, work disengagement, and career identity in Internet enterprises. The results show that avoidant job crafting reduces self-control resource depletion, curbing work disengagement, with career identity playing a moderating role. This research offers theoretical innovation by highlighting the positive side of avoidant job crafting and expanding the theory’s application. Practically, managers should view avoidant job crafting dialectically and guide employees. Employees should focus on valuable tasks, and enterprises can enhance employees’ career identity. Overall, these findings help employees and enterprises manage work stress and improve performance. However, this study has several limitations.

### Limitations and Future Research

There are some limitations to this study. First, the questionnaire data were collected in the same period of time and came from a single source platform, which may be influenced by the subjective factors of the subjects and a certain degree of self-section. Although we made efforts to minimize biases, no platform can avoid all biases. Therefore, the quality and effectiveness of the data need to be improved. Future studies can consider the combination of objective evaluation and subjective evaluation to measure each variable to improve the effectiveness of the data. At the same time, multi-period and multi-source data collection methods are adopted to improve data quality. Also, future studies could expand the channels for promoting the survey. Besides relying on the platform itself, researchers could collaborate with relevant industry associations, companies, and professional groups to disseminate the survey invitation more widely, thereby attracting a more diverse group of participants to further reduce bias. Second, we recognize that the small sample size for different hierarchical levels may have affected the comprehensiveness of our findings. Looking ahead, we believe that future studies could benefit from a larger and more balanced sample size, enabling a more detailed and meaningful comparison between employees at various hierarchical levels, which would contribute to a more comprehensive understanding of the relationships and phenomena under investigation. Thirdly, this study uses cross-sectional data, and the causality of time series between variables cannot be accurately inferred. In the future, longitudinal data on avoidant job crafting, self-control resource depletion, work disengagement, and career identity can be collected, and longitudinal research is considered to be carried out using longitudinal research design. In addition, the causality of time series between variables is deeply discussed by using experimental and time lag analysis methods. Fourthly, this study discusses the positive impact of avoidant job crafting from the perspective of resources combined with conservation of resources theory. Although it explores the impact of avoidant job crafting on work disengagement through self-control resource depletion, it does not reveal the working environment in which this impact is more effective. This study provides managers with a new idea to help employees reduce stress and get rid of negative work status. In the future, we can further explore which leadership style or work environment is more conducive to employees to reduce work disengagement through avoidant work crafting. Finally, currently, many industries are facing workplace competition and pressure. Due to the limitation of data collection, this study cannot collect data of employees in other industries to test the significance of this model in various industries. It is suggested that we should consider collecting employee data from other industries in the future, explore whether avoidant job crafting of employees in other industries has an impact on work disengagement, and test other mediating roles.

## Figures and Tables

**Figure 1 behavsci-15-00611-f001:**
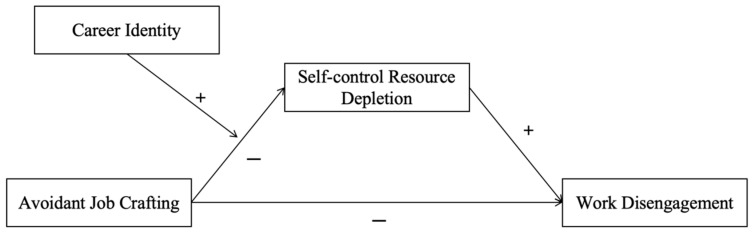
Theoretical model.

**Figure 2 behavsci-15-00611-f002:**
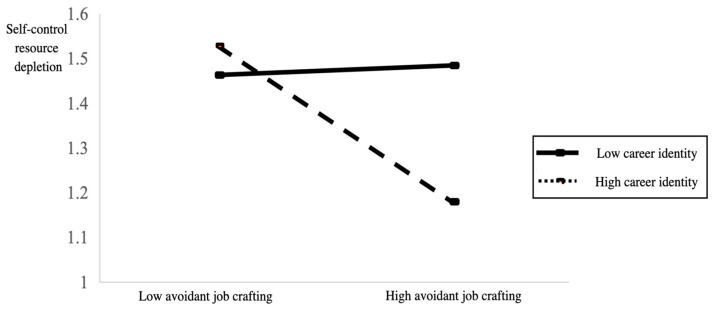
Moderation decomposition diagram of career identity on avoidant job crafting and work disengagement.

**Table 1 behavsci-15-00611-t001:** Harman single-factor test results.

Component	Initial Eigenvalues			Extracted Loading Squares		
	Total	Variance Percentage	Cumulative Percentage %	Total	Variance Percentage	Cumulative Percentage %
1	10.30	41.19	41.19	10.30	41.19	41.19
2	2.73	10.91	52.10	2.73	10.91	52.10
3	2.28	9.12	61.22	2.28	9.12	61.22
4	2.15	8.59	69.81	2.15	8.59	69.81
5	0.69	2.75	72.55			

Note: Extraction method is principal component analysis; sample size N = 455.

**Table 2 behavsci-15-00611-t002:** Summary of correlation coefficients, AVE square roots, AVE, and CR for various variables.

	1	2	3	4	AVE	CR
1. Avoidance job crafting	**0.77**				0.59	0.89
2. Self-control resource depletion	−0.32 ***	**0.83**			0.69	0.92
3. Work disengagement	−0.29 ***	0.24 ***	**0.77**		0.59	0.92
4. Career identity	0.40 ***	−0.27 ***	−0.25 ***	**0.84**	0.70	0.93

Note: Bold values on the diagonal represent the square root of AVE; *** *p* < 0.001.

**Table 3 behavsci-15-00611-t003:** Mean and standard deviation of each variables.

Items	Mean	Standard Deviation
Avoidant job crafting ([47])	3.49	0.90
1. I make sure that my work is mentally less intense	3.34	1.15
2. I try to ensure that my work is emotionally less intense	3.44	1.03
3. I manage my work so that I try to minimize contact with people whose problems affect me emotionally	3.49	1.11
4. I organize my work so as to minimize contact with people whose expectations are unrealistic	3.11	1.13
5. I try to ensure that I do not have to make many difficult decisions at work	3.71	1.05
6. I organize my work in such a way to make sure that I do not have to concentrate for too long a period at once	3.83	1.14
Self-control resource depletion ([32])	2.42	0.97
1. I feel drained	2.23	0.96
2. My mind feels unfocused right now	2.70	1.16
3. Right now, it would take a lot of effort for me to concentrate on something	2.33	1.07
4. My mental energy is running low	2.48	1.28
5. I feel like my willpower is gone	2.37	1.13
Work disengagement ([42])	2.46	0.83
1. I can always find something new and fun in my work (reversed)	2.44	0.88
2. I usually talk about my work in a derogatory way	2.85	1.07
3. Lately, I’ve been thinking very little about my work, just getting things done mechanically	2.34	1.18
4. I find my work challenging (reversed)	2.56	1.18
5. If this continues, I may gradually distance myself from my job	2.38	0.92
6. Sometimes, my work assignments make me feel uncomfortable	2.89	1.02
7. The type of work I do fits my identity (reversed)	2.13	1.07
8. I get more and more engaged in my work (reversed)	2.09	0.96
Career identity ([35])	3.58	0.94
1. When someone praises my career, it feels like a personal compliment	3.94	0.91
2. I am very interested in what others think about my career	3.42	0.96
3. When someone criticizes my career, it feels like a personal insult	3.29	1.23
4. When I talk about my career, I usually say ‘we’ rather than ‘they’	3.61	1.25
5. My career’s successes are my successes	3.65	0.99
6. If a story in the media criticized my career, I would feel embarrassed	3.58	1.17

**Table 4 behavsci-15-00611-t004:** Results of the hierarchical regression analysis.

	Self-Control Resource Depletion	Work Disengagement		
	Model 1	Model 2	Model 3	Model 4
Gender	−0.02	−0.03	0.01	−0.03
Age	0.06	0.03	−0.00	0.02
Education	−0.12 ***	−0.16 ***	−0.20 ***	−0.140 **
Tenure	−0.01	0.01	0.08	0.01
Position	0.02	−0.05	−0.04	−0.05
Job level	0.02	0.04	0.04	0.04
Avoidant job crafting	−0.08 *	−0.36 ***		−0.34 ***
Self-control resource depletion			0.27 ***	0.22 ***
R^2^	0.56 *	0.31 ***	0.25 ***	0.34 ***
ΔR^2^	0.00 *	0.10 ***	0.03 ***	0.02 ***

Note: *** *p* < 0.001; ** *p* < 0.01; * *p* < 0.05.

**Table 5 behavsci-15-00611-t005:** Results of the moderated mediation effect.

Career Identity		Moderated Indirect Effect				Moderated Mediation Effect			
		IND EFFECT	BootSE	BootLLCI	BootULCI	Index	BootSE	BootLLCI	BootULCI
Higher level (M + 1SD)	4.61	−0.03	0.01	−0.06	−0.01	−0.17	0.01	−0.34	−0.01
Average level (Mean)	3.67	−0.02	0.01	−0.03	0.00				
Lower level (M − 1SD)	2.72	0.00	0.01	−0.02	0.02				

Note: “Lower level” represents the value at 1 standard deviation below the mean, “Mean” represents the value at the mean, and “Higher level” represents the value at 1 standard deviation above the mean; “IND EFFECT” represents the indirect effect; BootSE represents the standard error; BootLLCI and BootULCI are the upper and lower limits of the 95% BC bootstrap confidence interval.

## Data Availability

The data supporting this study are available from the corresponding authors upon reasonable request.

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
