# Peer review of "“Avoidance” Is Not “Escape”: The Impact of Avoidant Job Crafting on Work Disengagement"

_behavsci, 2025, doi:10.3390/bs15050611_

Round 1

Reviewer 1 Report

Comments and Suggestions for Authors

The research article entitled “‘Avoidance’ Is Not ‘Escape’: The Impact of Avoidant Job Crafting on Work Disengagement” examines the impact of avoidant job crafting on work disengagement, as well as the mediating role of self-control depletion and the moderating role of career identity. The aim is to strengthen positive perceptions of avoidant job crafting among companies and researchers and to help employees and managers effectively utilize it to maintain good working conditions.

The topic of the research article is highly relevant, as it discusses "job crafting", a concept that appears frequently in recent publications. The authors specifically focus on "avoidant job crafting" and explore the conditions under which this behavior can positively impact organizations. Three hypotheses were formulated and confirmed in the study.

The authors conducted an analysis based on an online survey, evaluating a total of 455 correctly completed questionnaires. However, based on the information in the research article, I cannot assess the adequacy of the research sample. Additionally, the authors did not mention calculating the required sample size (482 questionnaires were distributed). The survey included 327 regular employees and 33 middle-level and above managers. The results do not specify whether there were differences between responses from employees in higher positions—a potential area for improvement (suggestion for the authors).

In the Discussion chapter, references should be included, as the current version lacks citations. The discussion is a crucial part of the manuscript, as it compares the authors' findings with previous research in the field.

Discrepancies with the article template:

  • The abstract exceeds 200 words, which does not comply with the template.
  • The research article is missing a “Conclusions” chapter. I recommend incorporating section 5.3. Limitations and Future Research into the Conclusions.
  • References need to be formatted according to the template. There are errors in lines 524–525.

Reviewer 2 Report

Comments and Suggestions for Authors

Dear author(s) of “Avoidance” Is Not “Escape”: The Impact of Avoidant Job Crafting on Work Disengagement. Thanks for providing this study, which I have read with interest. I have a few comments that hopefully might be useful in improving the presentation of the study.

  1. The introduction is difficult to understand and needs attention from a native English-speaking person to clearly communicate the background and the concepts used in this study. Several concepts are introduced without definitions, making the text unclear. Furthermore, a clear aim or research problem statement would help the reader understanding what this study will be about. This part must be re-written to make the reader able to understand what the study deals with and include figure 1 as support for the presentation. Examples are:
    Page 1-2, line 40 -45: The text is difficult to comprehend, and claims are stated without references as support. Please rewrite and develop this part.
    Line 47 has reads “have engaged in work disengagement” – please rewrite as this combination of words is confusing.
    Page 2, line, page 58: the phrase “and ensuring that goals are achieved” is unclear – whose goals are you referring to?
    Page 2, lines 70-72: The sentence “Paying too 70 much attention to employees’ negative states cannot fully understand the possible positive effects of avoidant job crafting” is unclear. Please rewrite.
    Page 2, lines 72-79: this is a very long sentence which is hard to understand. Please rewrite.
    Pages 2-3, lines 99-103: Again, this is difficult to read, please rewrite.
  2. Theory and hypotheses: This section suffers also from unclear language where words are lacking, or inadequate expression are used. A reader skilled in English should be of great value to make this more accessible for the reader. The arguments given for the hypotheses are rather unclear and lack convincing demonstrations of relationships between the involved concepts from previous studies.
    Page 3, lines 128 – 130: The explanation of given here is hard to understand.
    Page 3, lines 130-132: The statement “individuals are more sensitive» has no reference for comparison. This needs a rewrite.
    Page 4, lines 154-156: The first part of this sentence read like a definition, but the second part deals only with staying in one’s job, thus it loses meaning.

Page 4, lines 160-175: No references to previous research are given here to support the arguments for the hypothesis 3.

  1. The study:
    Page 5, line 188: reference lacking for the claim about data quality.
    Page 5, lines 183 – 195: As judged by respondents age, the sample is young and hardly representative.
    Page 6, lines 238 – 244: The example items of the Work disengagement scale and the Career identity scale are the same, and none of them are included in the full scales as presented in the supplementary material.

Discussion:
Page 11, lines 364 – 384: The way this part is written indicates that there are causal effects between independent and intervening/dependent variables. While this may be correct, this study leans on a cross-sectional design and such arguments are therefore not legitimized. This problem is acknowledged in the Limitation section and causality arguments should not be stated here.
 Page 11 -12, Section on Theoretical implications: the arguments here are hard to evaluate as there is absolutely no reference given in this text.

I am sorry for sending you all these critical comments, however, in its current state this paper does not meet the ordinary standards for being published. It has some potential if reworked into a clear and readable format. I wish you good luck with the undertaking. 

Comments on the Quality of English Language

The paper would profit greatly for involving in the re-writing a native English speaking scholar who also know this area of research. 

Round 2

Reviewer 2 Report

Comments and Suggestions for Authors

Dear author(s),

Thank you for providing an improved version of the manuscript. In my opinion, it is interesting and now close to publishable. I would recommend some few more improvements:

  1. There is an alternative line of research on work engagement led by the Deutch professor Wilmar Schaufeli. Although his scale was not used in the present study, his work should be referred to in relation to the engagement concept.
  2. An explanation of the self-control concept and a clear definition should be found in the present line 38 ff, as these theories are not generally known among the potential readership of this journal, and an understanding of this concept is needed to fully understand the study. You have a short definition of this concept in lines 131 ff, but this is not a sufficient explanation, and it should appear before introducing the self-control resource depletion in line 40.
  3. The reference style both in the text and in the reference list is inconsistent.
  4. Table 4, model 4: One of the regression coefficients is presented with three decimals, please reduce to two.

I wish you good luck!
